# Cell-to-Cell Interactions Mediating Functional Recovery after Stroke

**DOI:** 10.3390/cells10113050

**Published:** 2021-11-06

**Authors:** Claudia Alia, Daniele Cangi, Verediana Massa, Marco Salluzzo, Livia Vignozzi, Matteo Caleo, Cristina Spalletti

**Affiliations:** 1Neuroscience Institute, National Research Council (CNR), Via G. Moruzzi 1, 56124 Pisa, Italy; vera.massa@in.cnr.it (V.M.); marco.salluzzo@in.cnr.it (M.S.); caleo@in.cnr.it (M.C.); spalletti@in.cnr.it (C.S.); 2Department of Neurosciences, Psychology, Drugs and Child Health Area, School of Psychology, University of Florence, 50121 Florence, Italy; daniele.cangi@unifi.it; 3Department of Biomedical Sciences, University of Padua, Viale G. Colombo 3, 35121 Padua, Italy; livia.vignozzi@phd.unipd.it

**Keywords:** stroke, astrocytes, oligodendrocytes, microglia, interneurons

## Abstract

Ischemic damage in brain tissue triggers a cascade of molecular and structural plastic changes, thus influencing a wide range of cell-to-cell interactions. Understanding and manipulating this scenario of intercellular connections is the Holy Grail for post-stroke neurorehabilitation. Here, we discuss the main findings in the literature related to post-stroke alterations in cell-to-cell interactions, which may be either detrimental or supportive for functional recovery. We consider both neural and non-neural cells, starting from astrocytes and reactive astrogliosis and moving to the roles of the oligodendrocytes in the support of vulnerable neurons and sprouting inhibition. We discuss the controversial role of microglia in neural inflammation after injury and we conclude with the description of post-stroke alterations in pyramidal and GABAergic cells interactions. For all of these sections, we review not only the spontaneous evolution in cellular interactions after ischemic injury, but also the experimental strategies which have targeted these interactions and that are inspiring novel therapeutic strategies for clinical application.

## 1. Introduction

Currently, adult disability caused by stroke is still one of the major problems for public health organizations over the world. Indeed, the year 2017 saw 1.12 million of stroke episodes in the European Union, with 0.46 million deaths and 7.06 million suffering from disability, expected to increase by 27% by 2047 because of improved survival rates [1]. The 2021 Heart Disease & Stroke Statistical Update from the American Health Association states that in 2019, 6.6 million deaths were attributable to cerebrovascular disease worldwide and a total of 3.3 million individuals died of ischemic stroke [2]. Depending on the location, ischemic stroke can potentially induce impairments affecting cognitive, sensory and/or motor domains and, together with heart disease, represents 82% of disability-adjusted life years (DALYs) due to cardiovascular disease [3], implicating a large degree of assistance. In a narrow time window after ischemic stroke, administration of recombinant tissue plasminogen activator (tPA) and thrombectomy are effective strategies which improve recanalization and functional outcome. In the chronic phase, rehabilitation based on sensory-motor stimulations is one of the most used and important strategies to improve functional recovery [4].

Understanding cell-to-cell interactions, among different cellular counterparts, and molecular mechanisms mediating both neuroprotection and plasticity, is fundamental for designing effective pharmacological strategies to improve neural repair and widen the window of plasticity for functional recovery in the ischemic brain. When referring to stroke, several neuroprotective, anti-inflammatory and anti-apoptotic drugs were tested with the aim of limiting lesion size. Post-ischemic neuroprotective interventions were used to delay the time limit for thrombolytic intervention in rodent models. Those interventions may involve immunosuppressive agents [5], mediators of inflammatory response [6] and NMDAR-mediated NO production [7], but also stem cells [8] and hypothermia [9]. Other novel approaches aimed at targeting more specific factors of poor prognosis in stroke patients, like those regulating water homeostasis and glymphatic clearance in brain tissues (i.e., aquaporin channels) [10,11,12]. Unfortunately, according to the American Heart the American Stroke Association Guidelines for the Early Management of Patients with Acute Ischemic Stroke, up to now there are neither pharmacological nor non-pharmacological treatments recommended as neuroprotective agents [13]. This evidence suggests the necessity to go deeper inside the substrate of post stroke plastic rearrangements to identify more effective cellular and molecular targets. Perilesional tissue, in particular, is the theatre of a fundamental reorganization that can be the key for an improved functional recovery, if properly guided. This reorganization involves almost the entire variety of cell populations in the brain tissue, including excitatory and inhibitory neurons and glial cells. All of these actors react to the infarct with a massive morpho-functional reorganization and secrete specific neurochemical signals that change their interactions with nearby cells. Understanding the consequences of these interactions and finding how to manipulate them is an existing challenge for post stroke therapy.

In this review, we present a summary of the principal cellular players that mediate neuroprotection after brain ischemia or, instead, have a toxic effect in the damaged tissue. In particular, we focus on cell-to-cell interactions involving astrocytes, oligodendrocytes, microglia, neurovascular components and pyramidal cells/interneurons, mediating support, plasticity or toxic effects to neurons in the damaged brain.

## 2. Cellular and Molecular Support from the Neurovascular Unit after Ischemia

The neurovascular unit (NVU) gathers together all the cellular and extracellular components that are responsible for the maintenance of the blood brain barrier (BBB) selectivity, cerebral homeostasis, as well as the control of cerebral blood flow [14]. The NVU incorporates neurons, astrocytes, endothelial cells (ECs), pericytes (PCs), microglia, and basement membrane. Each component shares intimate and reciprocal association to each other, establishing an anatomical and functional whole, which results in a very efficient system of cerebral homeostasis and blood flow regulation [15,16].

Additionally, NVU has roles in physiological conditions; increasing evidence showed that vascular lineage components of the NVU (endothelial cells and PCs) have the potential to promote post-stroke brain repair through neuroprotection, neural regeneration and by the formation of the neurovascular niche [17].

Guo et al. demonstrated that vascular endothelial cells are capable to protect primary cortical neurons against oxygen-glucose deprivation, oxidative damage, endoplasmic reticulum stress, hypoxia, and amyloid neurotoxicity in vitro through BDNF secretion [18].

Moreover, in response to shear stress of the blood flow, ECs upregulate expression of the neuroprotective molecule osteopontin, resulting in ischemic tolerance acquirement [19].

Furthermore, ECs showed the ability to promote BBB reconstruction directly by increasing VE-cadherin expression after an ischemic insult in adult mice [20] and indirectly by the recruitment of PCs expressing platelet derived growth factor receptor β (PDGFRβ) in the MCAO model [21,22]. Moreover, in the acute phase, PDGFRβ positive PCs extend from peri-infarct areas toward the ischemic core, secreting collagen type I and fibronectin, resulting in fibrosis and reduction of the infarct area [23,24].

Interestingly, PCs seem also to be capable to acquire stem cells multipotency through reprogramming [25] and differentiate into various cells including neural and vascular cells [26,27,28,29,30]; indeed PCs extracted from ischemic regions of mouse brains and human brain PCs cultured under oxygen/glucose deprivation develop stemness in order to replace major components of the NVU [31].

Nevertheless, it is known that ECs have a key role in vascular niches of the microenvironment of conventional neurogenic zones [32,33], such as the subventicolar zone (SVZ) [34] and the sub granular zone (SGZ) [35]. In fact, the murine ECs line stimulate embryonic-derived neural stem cells self-renewal when cocultured in vitro [36], while cerebral ECs activated by ischemia (isolated from the stroke boundary) promote neuronal differentiation and reduce astrocytic differentiation in SVZ neural progenitors [37].

A recent paper showed that neural stem/progenitor cells (NSPCs) could also originate directly in the ischemic area of the adult murine brain, and can differentiate into electrophysiologically functional neurons, astrocytes and myelin-producing oligodendrocytes, participating to the post-stroke cortical reconstruction [38]. As reported by Nakagomi et al., ECs increase survival, proliferation and neuronal differentiation of ischemia-induced NSPC when cotrasplanted with cortex-derived stroke-induced NSPC, onto adult mice that have undergone MCAO [39]. In addition, transplanted bone marrow mononuclear cells (BMMCs) seem to promote neurogenesis and functional recovery in the MCAO mouse model through the proliferation of ECs, since treatments with endostatin (known to inhibit ECs proliferation), following BMMCs transplantation, suppress proliferation of NSPCs, neurogenesis and functional recovery [40].

Finally, while mature neurons and glia are very sensitive to ischemic insults, ECs and PCs can survive for several days even within the ischemic area [20,41]. This could be crucial for future therapies, since a lethal ischemia with neural cell death but without vascular cell death (obtained in mice by an early reperfusion following 90 min MCAO), strongly increases the healing process, neurogenesis, gliogenesis and functional recovery compared to a lethal ischemia with both neural and vascular cell death (permanent MCAO) [42,43].

In conclusion, although the precise roles of vascular cells that survive within ischemic areas in post-stroke conditions remains unclear, ECs and PCs seem to have the potential to promote brain repair through several mechanisms.

## 3. Pathophysiological Role of Astrocytes and Reactive Gliosis after Brain Ischemia

An ischemic stroke is a devastating metabolic shock for brain tissue and potently impacts not only on neuronal populations, but also on the organization of their supporting system, i.e., glial cells and particularly astrocytes. The adult brain is responsible for the consumption of 20% of oxygen (O2) and 20–25% of glucose utilization [44,45]. Most of neuronal energy consumption takes place at the synaptic level, where it was postulated that astrocytes play a role in the neuronal metabolism of glucose. According to the astrocyte-neuron lactate shuttle (ANLS) hypothesis postulated in 1994 [46], when neuronal synaptic activity intensifies, astrocytes raise the rate of glucose uptake from the blood stream, activating the glycolytic pathway and the lactate production. This substrate, once released in the extracellular space, is used by neurons as an alternative energetic source [46]. According to this, astrocytes serve as a ‘lactate source’ whereas neurons serve as a ‘lactate sink’ [47]. The ANLS theory has been challenged by Bak and colleagues, who alternatively proposed that the metabolism of lactate in neurons is coupled to the activity of the malate-aspartate shuttle (MAS), an enzyme involved in the reduction of the NADH produced by glycolysis. MAS activity is limited by the raise in intracellular Ca^2+^ following depolarization, suggesting that lactate metabolism can take place only during repolarization (and in the period between repolarization), instead of during neurotransmission activity as stated by the previous theory [47,48]. However, numerous pieces of evidence in support of the ANLS hypothesis have led to the wide acceptance of this theory [47].

In the healthy cortex, astrocytes also play a critical role in the brain physiology, such as maintaining ionic homeostasis [49], mediating functional hyperemia [50], removing glutamate released in the synaptic cleft to avoid excitotoxicity [51], and contributing to neuronal electrical activity through stimulation of different G-protein coupled receptors (GPCRs: [52,53,54]). Moreover, these cells participate with the ‘tripartite’ synapse, where they can regulate gliotransmitters release in the synapses after an increase of intracellular Ca^2+^ levels in response to neurotransmitters [55,56].

After the onset of ischemia, the physiological functions of astrocytes begin to be altered, eliciting several morphological and biochemical modifications which have a severe impact on the brain region affected by stroke. For instance, it was demonstrated that after ischemia astrocytes contribute to release glutamate in the synaptic cleft, due to transporters running in reverse, leading to extracellular glutamate elevations and severe excitotoxicity [57,58]. Moreover, one of the first astrocytic reactions to ischemic damage is represented by enhanced Ca^2+^ signaling [59]. Indeed, by means of calcium imaging in brain slices, it has been found that calcium activity in astrocytes is quiescent under normal conditions, but frequent Ca^2+^ elevations are detectable in an oxygen-glucose deprivation (OGD) model [60]. In the same work, the authors showed that OGD induced slow inward currents (SICs) mediated by extrasynaptic NMDA receptors in CA1 pyramidal neurons. In this way, enhanced Ca^2+^ activity in the astrocytic network plays a key role in the activation of extrasynaptic NMDA receptors in hippocampal neurons, heightening glutamatergic signaling and brain damage during ischemia. Similar results were also found in an in vivo setting: using in vivo two-photon imaging, Ding et al. (2009) found that astrocytes exhibit intercellular Ca^2+^ waves starting 20 min after a photothrombosis (PT)-induced ischemia. The magnitude of theCa^2+^ signal was greater in the ischemic core than in the perilesional region, showing a different spatial activation of astrocytes in the lesioned region. Notably, inhibition of astrocytic Ca^2+^ signal with BAPTA Ca^2+^ chelator reduced infarct volume, suggesting that the increased Ca^2+^ signal in astrocytes contributes to ischemic damage [61].

Beyond enhanced Ca^2+^ signaling, it was shown that astrocytes are also involved in the development of edema following a traumatic brain injury or a stroke. Water homeostasis in the brain is regulated by a family of cellular membrane proteins called aquaporins (AQP) [62,63], of which the AQP4 is the principal isoform expressed in the CNS, abundant especially in astrocytes [64,65]. Recently, it has been demonstrated that brain and spinal cord edema is associated with increases in AQP4 expression in astrocytes and with their subcellular translocation to the blood-spinal-cord-barrier (BSCB), raising the flux of water in astrocytes [11]. Moreover, the same authors demonstrated that AQP4 membrane localization is mediated by a calmodulin (CaM) and protein kinase A (PKA) mechanism. The use of trifluoperazine (TFP), a CaM antagonist, helped in reducing edema in a rat spinal cord injury model, promoting functional recovery of the sensory and locomotor deficit following the injury [11]. This role was recently confirmed also in a photothrombotic stroke mouse model, where authors showed that administration of TFP during the early acute phase of stroke effectively reduced the cerebral edema and the AQP4 mRNA and protein expression levels in the brain [12]. Moreover, using spectroscopy and X-ray fluorescence imaging, authors showed that TFP significantly increases the level of glycogen in the peri-infarct tissue, which may have a neuroprotective effect by providing supplemental metabolic energy in the acute post-stroke phase [12].

The acute response of astrocytes comprises also morphological changes associated with upregulation on the levels of the glial fibrillary acidic protein (GFAP), a mechanism referred to as reactive astrogliosis [66,67,68,69]. These dynamic modifications of reactive astrocytes were measured in Mestriner et al. (2011), where the authors used immunohistochemistry for GFAP protein in the penumbra region 30 days after ischemic and hemorrhagic stroke in rats. Results showed an increased optical density of GFAP-positive astrocytes after stroke and increased complexity of primary processes, measured as the number and length of ramifications, compared to a sham control group [67]. A similar analysis was performed by Wagner et al. (2013), where reactive gliosis was investigated 4 days after MCAO in spontaneously hypertensive rats. Measurement of the GFAP staining in several brain regions revealed that in the penumbral region the volume, diameter, length, and branching of processes of reactive astrocytes were increased compared with the astrocytes in the contralateral hemisphere and in remote regions away from the ischemic core. These data indicate a regionalization of reactive gliosis, depending upon the distance from the site of brain injury [68].

Moreover, after ischemic stroke reactive astrocytes in the peri-infarct region form a glial scar in the penumbra that demarcates the ischemic core from healthy tissue. Progression of reactive gliosis and glial scar formation was described in detail by Li et al. (2014), where authors examined GFAP expression in a mouse model of photothrombotic ischemia at different time points after the induction of the lesion. The formation of the glial scar starts at 6 days post stroke (dps), when reactive astrocytes shift their morphology from the hypertrophic shape of the acute phase (observed from 1 to 4 dps) to a thinner shape with elongated processes directed towards the ischemic core. After 10 dps, reactive astrocytes in the perilesional region were stable, indicating complete maturation of the glial scar. In the same study, the authors documented that gliosis is also characterized by proliferation of a fraction of reactive astrocytes in the perilesional region. Using bromodeoxyuridine (BrdU) labeling and immunostaining, they observed that the percentage of BrdU+ reactive astrocytes significantly increased in the acute phase after PT, reaching their peak at 4 dps. Finally, they also examined the functional deficit after ischemia through several sensory-motor tasks: Schallert cylinder, hanging wire, pole, and adhesive removal tests. The largest functional deficits occurred from days 2 to 4 after ischemia and significant functional recovery starts after day 6 post stroke [69].

Following stroke, reactive astrogliosis has a controversial functional role since it may participate in brain repair but could also limit neuronal outgrowth and recovery after the lesion. Indeed, reactive astrocytes express a broad range of molecules that are inhibitory for axonal regeneration, such as chondroitin sulfate proteoglycans (CSPGs) [70,71,72], thus reducing neuroplasticity in the cortical tissue after ischemia. Along this line, a study showed that ephrin-A5 is induced in reactive astrocytes in the peri-infarct cortex and is an inhibitor of axonal sprouting and motor recovery in stroke in motor areas [73]. Indeed, pharmacological blockade of ephrin-A5 signaling determined robust sprouting of axonal connections in motor, premotor and prefrontal perilesional regions, and mediates a significant improvement in recovery of forelimb motor function as assessed by behavioral tests (i.e., grid walk and Schallert cylinder tests). Interestingly, effects on axonal sprouting were more evident when the blockade of ephrin-A5 was combined with the forced use of the affected forelimb (via delivery of the synaptic blocker botulinum neurotoxin into the ipsilesional forelimb) [73]. However, it was shown that mice double-knockout for two major astrocytic intermediate filament proteins, glial fibrillary acidic protein (GFAP) and vimentin (GFAP–/–Vimentin–/–mice), develop less dense scars around the injury site but they paradoxically display reduced motor recovery and axonal remodelling of corticospinal fibers [71]. In another report, Hayakawa et al. (2010) used fluorocitrate, a metabolic inhibitor of astrocytes, to reduce reactive gliosis. They showed a worsening of the behavioral deficits in the treated animals, consistent with a restorative effect of the glial scar [74].

Thus, reactive astrocytes clearly play a dual role post-stroke, as on one hand, they inhibit sprouting and axonal growth, but they may also participate in repair by producing or recycling neurotrophic factors, thus stimulating plasticity of spared networks. The glial scar may also physically isolate the injury site from viable tissue, preventing a cascading wave of uncontrolled tissue damage [75], and restrict diffusible factors secreted from the damaged region.

Altogether, the available literature presents controversial data regarding the role played by astrocytes in post-stroke functional recovery. Some reports indicate a block of sprouting and plasticity by reactive astrocytes while other studies support the view that astrocytes may play a role in aiding functional restoration after stroke.

## 4. Role of Oligodendrocytes in Neural Support and Sprouting Inhibition

White matter (WM) myelin sheaths insulate axons, facilitating conduction of action potentials and increasing bioenergetical efficiency [44]. In the central nervous system, WM is formed and maintained by oligodendrocytes (OLGs) [76].

OLGs are highly susceptible to oxidative stress, trophic factors deprivation and glutamate activity (Arai 2009), therefore WM lesions are a frequent consequence of ischemic events in stroke patients and experimental animal models of cerebral ischemia [77,78].

Despite OLGs not proliferating in the adult [79], oligodendrocyte precursor cells (OPC) may enable myelin sheath renewal by differentiating into oligodendrocytes throughout all life [80]. After brain injury involving the WM, OPC proliferate in the SVZ [81] and migrate to the perilesional area to become a myelinating OLG, attempting to restore the damage [82,83,84,85]. This process of post-stroke oligodendrogenesis is influenced by complex interactions among several cell populations, thus offering multiple targets for therapeutic interventions aimed at increasing the recruitment of OPCs to enhance white matter repair after injury.

Neuronal activity appears to have a direct role in regulating OPCs proliferation and differentiation [86] and some patterns of activity are more likely to promote proliferation, while others are more likely to promote differentiation [87]. Interestingly, OPCs receive both excitatory and inhibitory synaptic inputs from neurons, which modulate several pathophysiological processes. Glutamatergic synaptic contacts between neurons and OPCs are mediated by AMPA receptors (AMPARs), and Ca^2+^ permeability of AMPARs at these synapses could be a key mechanism in modulating the development of oligodendroglial cells. Indeed, inducing expression of AMPARs with different Ca^2+^ permeability in mouse OPCs of corpus callosum during the peak of myelination deeply affects OPC proliferation and differentiation [88]. Moreover, it seems that OPCs differentiation could be induced directly by demyelinated neurons, as showed by the enhanced differentiation of OPCs into myelinating OLGs, after recurrent or moderate optogenetic stimulation of neurons in the corpus callosum in a mouse model of focal lesion. These results highlight the importance of neuronal-OLG interaction even during post stroke recovery [89].

OLGs were proved to also have neuroprotective activity, as shown by the ability of OLGs to sustain callosal axons, through gap junctions, after exogenous glucose deprivation in ex-vivo brain slices [90]. Interestingly, even OPCs seems to have a neuroprotective role, through secretion of the insulin-like growth factor-1 (IGF-1). Indeed, medium derived from OPC cultures increase in vitro neuronal survival, while this effect is blocked by neutralizing IGF-1 [91].

However, mature OLGs are predominantly considered impediments to post-stroke neural regeneration. In 1988 Schwab and Caroni identified OLGs and the white matter as non-permissive substrates for neurite outgrowth [92]. The molecules responsible for this neurite outgrowth inhibition are collectively termed Myelin associated inhibitors (MAIs) and include Myelin Associated Glycoprotein (MAG), Oligodendrocyte Myelin glycoprotein (OMgp) and Nogo A. These ligands mainly act through three primary receptors: Nogo receptor 1 (NgR1), paired immunoglobulin-like receptor B (PirB) and Sphingosine-1-phosphate receptor 2 (S1PR2) [93,94,95] and are generally associated with axonal growth cones collapse and repair inhibition following CNS injury.

In fact, neutralizing Nogo-A in the presence of a middle cerebral artery occlusion (MCAO) in rats results in enhanced cortico-efferent projections and functional improvements [96,97]. Interestingly, administration of a monoclonal antibody against Nogo-A after an ischemic lesion increases dendritic arborisation, suggesting a role of this molecule also in limiting dendritic plasticity after stroke [98]. Exogenous inhibition of NgR1 with peptide antagonists was shown to mitigate axonal damage, enhance axonal sprouting and improve motor function following cortical injury in rodents [99]. Moreover, the use of a NgR1 decoy protein [100] showed efficacy in the recovery of rats subjected to MCAO, when administered 1 week post-injury for a period of 28 days [101]. Finally, a combination of Nogo-A neutralization followed by rehabilitative training revealed an almost complete restoration of skilled forelimb functions in rats with large photothrombotic stroke, due to an extensive and precise reinnervation of the stroke-denervated spinal hemicord by midline-crossing fibers from the intact motor cortex and corticospinal tract [102].

Overall, these studies underline the critical role of OLGs, and the importance of OPCs maturation timing to avoid neurite outgrowth inhibition during stroke recovery.

As previously mentioned, OLGs and OPCs can offer neuroprotection and metabolic support to vulnerable neurons, and indeed other cell types providing support to OLGs and OPCs can be considered potential therapeutic targets for post-stroke neuroprotection.

In this context, astrocytes are known to support OLGs functions through gap junctions [103] and protect OPCs from oxidative stress, starvation and hypoxia throughout their secretome [104]. A main actor in this process is Erythropoietin (EPO), a glycoproteic cytokine showing in vitro protective effect on OPCs subjected to hypoxia-reoxygenation injury [105]. Moreover, administration of EPO following MCAO in rats results in increased neurogenesis and oligodendrogenesis [106].

Additionally, astrocytes increase in vitro maturation of OPCs, subjected to hypoxia, via a BDNF-dependent mechanism [107]. On the contrary, transgenic mice with reduced expression of BDNF from reactive astrocytes undergo increased damage and less myelination following carotid stenosis [107]. Accordingly, post-stroke intravenous administration of BDNF in rats increases oligodendrogenesis, remyelination and recovery after 4 weeks from the treatment [108]. Finally, astrocytes react to an ischemic insult by secreting Leukemia inhibitory factor (LIF), a protein that was found to promote OLGs survival and functional recovery after MCAO [109].

OLGs are also supported by endothelial cells through trophic factors secretion, as demonstrated in-vitro [110]. Endothelial secretome also enhance endothelial and OPCs proliferation and potentiate OPCs maturation [111]. Furthermore, it increases vascular density, myelination and OLGs number, improving functional recovery after carotid stenosis in mice [111].

Conversely, secretome of hypoxic OPCs increases tubular formation of endothelial cells in vitro, and improves functional recovery and angiogenesis following MCAO in mice [112]. However, it seems that Nogo-A plays a crucial role in the inhibition of post-stroke vascularisation. Indeed, genetically deleting Nogo-A or its S1PR2 receptor results in increased vascular spouting and in neurological deficit reduction following photothrombotic ischemia [113].

## 5. Microglial-Mediated Inflammation in Stroke: A Double-Edged Sword

Microglia are considered the resident macrophages of the central nervous system and are the main effectors of brain immune function. Microglia are constantly sampling the brain in search of damaged neurons and infectious agents. When pathogens cross the blood-brain barrier, microglia cells react rapidly to increase inflammation, destroying foreign agents before they can damage tissue. On the basis of their vital functions in regulating neuroinflammation, microglia are an important target for stroke therapy. Neuroinflammation and microglial responses are involved in all phases of the ischemic cascade, from the acute event, which leads to the first wave of neuronal cell death, to the later stages involving phagocytosis and tissue remodelling.

Several pieces of evidence show that the immune response influences reparative mechanisms in the damaged brain. The influence of inflammation on adult neural stem cell regulation and function has also received much attention. Although the details of immune signaling in the central nervous system are not completely clear, it is known that the impact of inflammatory signaling on adult neurogenesis is focused on the activation of microglia as a source of proinflammatory cytokines, such as TNF-α, IL-6, and IL- 1β. Studies have shown that neural stem cells undergo apoptosis by TNF-α in vitro, suggesting that TNF-α has a negative effect on post-stroke neurogenesis (Iosif et al., 2008). Glucocorticoid-induced tumor necrosis factor (TNF) receptor (GITR), a multifaceted regulator of immunity belonging to the TNF receptor superfamily, is expressed on activated CD4^+^ T cells. Furthermore, GITR and its ligand GITRL are functionally expressed on brain microglia and it was shown that stimulation of GITRL can induce inflammatory activation of microglia [114].

Studies in a mouse model of cortical infarction have shown that GITR, triggering on CD4^+^ T cells, increases post-stroke inflammation and decreases the number of neural stem/progenitor cells induced by ischemia (iNSPCs). CD4^+^ and GITR^+^ T cells were preferentially accumulated at the post-ischemic cortex and mice treated with GITR-stimulating antibodies had increased post-stroke inflammatory responses with increased apoptosis of iNSPCs. In contrast, blockade of the GITR-GITR ligand (GITRL) interaction abolished inflammation and suppressed apoptosis of iNSPCs. These observations indicate that CD4^+^ T cells and GITR are the main modulators of post-stroke neurogenesis impairment. This suggests that blocking the GITR-GITRL interaction may be a novel immune-based therapy in stroke [115].

Microglia can also regulate post-stroke plasticity through other mediators such as microglia-neuron interactions mediated by the neural factors CD200 and CX3CL1.

CD200 glycoprotein is expressed primarily by neurons and its receptor, CD200R, is expressed on myeloid cells, including microglia. This interaction is involved in maintaining microglial cells in a quiescent homeostatic state. CD200 expression has been shown to be abundant in the healthy brain and in the contralesional hemisphere after ischemic injury. After transient middle cerebral artery occlusion (tMCAO), there is a dramatic decrease in CD200 levels that are re-increased at 7 and 14 days after reperfusion [116].

The acute role of CD200 has been studied in the first 48 h, in a mouse model of middle cerebral artery occlusion (pMCAO). Loss of neuronal CD200 contributes to microglia activation and associated neuronal death. Moreover, intracerebroventricular injections of CD200, performed after induction of pMCAO, reduced microglia activation and expression of cytokines TNF, IL-1β, and IL-10 [117]. A recent study assessed changes in monocyte infiltration, microglia proliferation, and behavioral deficits up to one week after injury in CD200R knock-out (CD200R-KO) mice subjected to transient MCAO. Increased monocyte and microglia infiltration was observed in CD200R-KO mice but no difference in lesion volume at 72 h after ischemia was reported, compared to control mice. Moreover, an increased mortality rate in the first week was observed in CD200R-KO mice, suggesting a role of the aggravated and prolonged inflammatory response, found up to 7 days in CD200R-KO but not in control mice, regardless of lesion volume. In addition to this, motor and behavioral assessment 7 days after injury showed a worsened performance in CD200R-KO mice compared to controls [118].

The role of CD200-DC200R interaction in post-stroke functional recovery has been recently supported by Sun and colleagues [119], who confirmed that CD200/CD200R signaling pathway contributes to spontaneous functional recovery in rats subjected to MCAO. Authors showed that post-stroke intracerebroventricular injection of CD200 (as an agonist of CD200R) improved sensorimotor function in a battery of behavioral tests: Longa test, adhesive removal test, limb-use asymmetry test and the modified grip-traction test. Better performances correlated with an enhanced synaptic plasticity, i.e., recovered density and morphology of dendritic spines, through inhibition of microglia activation and inflammatory factor release. In contrast, rats injected with a CD200R blocking antibody showed an aggravated sensorimotorfunction, accompanied by enhancedmicroglia activation and release of pro-inflammatory factors.

Thus, it appears that modulation of microglia function may be an effective tool for treatment of stroke. Bone marrow mesenchymal cells (BM-MSCs) have immune modulatory properties in the brain and could play an important role in regulating microglial cell activation during the acute phase of stroke. As demonstrated by Li et al., (2019), ischemic rats that underwent BM-MSC transplantation, 12 h after MCAO, showed a reduction in microglial activity 3 days after injury. Transplantation prevented apoptosis in peri-infarct neurons, confirming that BM-MSC transplantation effectively protects from acute neural degeneration [120]. Moreover, Kong et al. (2018) demonstrated that mesenchymal stem cells derived from the amniotic membrane of the human placenta, transplanted into ischemic rats, reduced the level of microglia activation as well as the level of pro-inflammatory cytokines and increased the expression of CD200. Post-stroke function was improved 1 week after lesion, remaining stable up to 6 weeks post-stroke in treated animals. Moreover, at this chronic time point the extent of injury was reduced in transplanted animals compared with the control group [121].

In the central nervous system CX3CL1 is expressed primarily by neurons, and its unique receptor CX3CR1 is expressed exclusively by microglia. The CX3CL1-CX3CR1 signaling pathway modulates microglia activation and regulates several microglial cell functions in the adult brain, influencing synaptic transmission both under pathological and physiological conditions [122]. Thus, CX3CL1-CX3CR1-mediated microglia-neuron interaction in experimental ischemic models was investigated as a target for neuroprotection.

CX3CR1 knockout mice undergoing focal cerebral ischemia have reduced IL-1β and TNF production, along with reduced ischemic volume, improved functional recovery, and less neural cell death [123]. Further studies confirmed that CX3CR1 deficiency may facilitate alternative microglial cell activation after stroke, suggesting that CX3CR1 abolition attenuates microglia proliferation and inflammatory capacity, improving neurological recovery [124]. There was also a reduction in the number of apoptotic neurons in CX3CR1-deficient mice in the infarct region, indicating that this receptor mediates cell death in ischemia [125]. Consistently, recovery of neurological function after ischemic injury can be rapidly improved by inhibition of the CX3CL1-CX3CR1 pathway [126]. Moreover, Tang and colleagues demonstrated that microglia are stationary within the lesion and that the lack of the CX3CR1 receptor prevents CX3CL1 from maintaining a low ramification number, i.e., a low level of microglia activity, indicating that CXRCL1 is a key factor in the induction of microglia from their amoeboid to hypertrophic form [127]. Therefore, studies with CX3CL1-deficient mice after transient MCAO, showed reduced ischemic area and reduced mortality when CX3CL1-CX3CR1 signaling was absent [128]. However, other studies support a protective effect of CX3CL1 in stroke. Indeed, CX3CL1 can also provide a neurotrophic effect through microglia-derived protective factors, including adenosine, with the activation of adenosine A1 receptors (A1Rs) which show an inhibitory effect on microglia activation [129,130]. In fact, intracerebroventricular administration of exogenous CX3CL1 to naive rats before pMCAO, reduces cerebral infarct size and neurological deficits, in a A1R-dependent manner. Indeed, in the presence of A1R antagonist, the neuroprotective effect of CX3CL1 pre-treatment on pMCAO was abolished. However, CX3CL1-induced neuroprotection is ineffective in CX3CR1- or CX3CL1-deficient mice, confirming the duality and complexity of microglia and inflammation in brain injuries [131].

In a clinical study, researchers hypothesized that patients with higher plasmatic levels of CX3CL1 after stroke would have a more robust inflammatory response and would have worse functional outcomes. Post-stroke immune responses through CX3CL1 levels were assessed from day 1 to day 180 in a cohort of 85 patients. Contrary to the original hypothesis, patients with better clinical outcome 6 months post-stroke had higher levels of CX3CL1 in blood plasma [132].

Altogether, CD200-CD200R and CX3CL1-CX3CR1 signaling appear to play an important dual and controversial role in neuroprotection and functional recovery; the ability to understand and pharmaceutically manipulate these pathways in an appropriate time window post-infarct could reduce neurological sequelae after stroke.

## 6. Pyramidal and GABA-Ergic Neural Interactions in Post-Stroke Plasticity

The ischemic event is known to trigger spontaneous and activity-dependent plastic rearrangements of neural connections across a great variety of cell populations, including pyramidal neurons and several subtypes of inhibitory interneurons.

Longitudinal in vivo two-photon imaging studies, targeting pyramidal neurons in somatosensory cortex, have shown that apical dendrites in perilesional tissue, in the first two weeks after stroke, display a spontaneous threefold increase in structural remodelling [133]. However, without any intervention, the shortening of tips detected closer to the infarct counterbalance the dendrite extension and the net length of dendritic arbour remained stable. This demonstrates that cortical pyramidal neurons preserve a plastic potential for important structural changes that could possibly be targeted, encouraged and guided by neurorehabilitative treatments. Indeed, we found that a combination of motor rehabilitation with neuroplastic intervention can decrease synaptic deletion and increase spine density in perilesional tissue in a mouse model of cortical stroke [134].

Fast-spiking, parvalbumin-positive (PV+) GABAergic interneurons exert a powerful control of cortical activity and evidence indicates a role of this neuronal population in post-stroke recovery.

Electrophysiological and optogenetic investigations targeting these cells have shown a clear deficit in synaptic transmission after brain ischemia [135]. Specifically, optogenetically evoked PV+ excitability was suppressed in the forelimb somatosensory cortex after five minutes common carotid arterial occlusion (CCAO), but recovered rapidly with reperfusion, while PV+ stimulation-evoked GABAergic synaptic network activity exhibited a prolonged suppression even ∼1 h after reperfusion. This suppression may be caused by the downregulation of postsynaptic GABAergic receptors, by the depression of presynaptic release or by the reversal of GABA transporter. These alterations were in line with the evolution of the functional deficit and demonstrate how the alteration of the inhibitory network after a stroke can influence the recovery of the perilesional tissue [136]. Moreover, the susceptibility of the inhibitory and the excitatory networks to post-stroke metabolic stress is not equal. Current-clamp recordings in prefrontal cortex slices exposed to oxygen-glucose deprivation and reoxygenation showed that PV-positive interneurons were more vulnerable to ischemic damage than pyramidal neurons, as indicated by the lower percentage of recovery of PV-positive interneurons. Specifically, large amplitude, presumably action-potential dependent, spontaneous postsynaptic inhibitory currents recorded from pyramidal neurons were less frequent after oxygen deprivation than in the control condition, while disynaptic inhibitory postsynaptic currents (dIPSC, recorded on Pyramidal Neurons after bipolar stimulation on the border of white matter and layer 6) in pyramidal neurons produced predominantly by PV+ interneurons were reduced [137]. Moreover, dendrites of PV-positive interneurons exhibited more pathological beading than those of pyramidal neurons. In addition, short-term in vitro ischemia, in rat brain slices, was shown to permanently impair the excitability of inhibitory neurons and synaptic transmission mediated by γ-aminobutyric acid (GABA), while principal neurons appear to be even more excitable during the reperfusion [138].

Parvalbumin interneurons request high energy levels to preserve their fast spiking action potentials [139]. This high energy demand leads to greater susceptibility to metabolic and oxidative stress [140,141]. PV+-interneurons have critical roles in the generation and maintenance of gamma oscillations, an additional energy requiring process [142]. Disruption of gamma oscillations have been linked to hypoxia and decrease of PV+ interneurons [143,144].

These data suggest a differential vulnerability to ischemic conditions of excitatory and inhibitory neurons, leading to the altered excitation-inhibition balance associated with stroke pathophysiology. The restoration of this balance is therefore a promising target for post-stroke neurorehabilitation as we recently demonstrated in mice [134,145] and could inspire novel therapeutic strategies with non-invasive neuromodulation techniques (Non-invasive brain stimulation, NIBS).

The evidence that GABAergic interneurons, despite being subject to functional alterations, survive the injury in several perilesional brain lesion, as demonstrated in a photothrombotic stroke model [146], demonstrate that acting on inhibitory interneurons as therapeutic strategy is a feasible solution.

Some studies exploited direct manipulation of interneurons activity to improve functional recovery after ischemic injury using electrophysiological, optogenetic and chemogenetic strategies. A recent paper demonstrated beneficial effect of inhibitory neurons stimulation in gamma (40 Hz) range in the acute phase after stroke in mice. The authors found reduced lesion volume and improved motor function, suggesting a neuroprotective effect of the treatment [147]. A different approach, targeting excitatory neurons was recently followed by Wang et al. which used chemogenetic techniques to selectively inhibit forebrain excitatory neurons in a transient MCAO ischemic model. Mice treated with Clozapine-N-oxide (CNO) which activated inhibitory DREADD receptors expressed by excitatory neurons, exhibited significantly improved neurologic function and smaller infarct volumes [148].

In addition, the cortex is not the only target for the modulation of GABAergic neurons to promote functional recovery after stroke. It was proved, in a murine tMCAo model, that optogenetic inhibition of striatal GABAergic activity was able to improve functional recovery, and reduced brain atrophy volume and cell death compared with the control, while activation of striatal GABAergic neurons resulted in adverse effects [149]. The positive effect is mediated by the upregulation of the basic fibroblast growth factor (bFGF) in endothelial cells, possibly orchestrated by astrocytes. This demonstrates that a precise instead of overall regulation of the excitatory/inhibitory system depending on the brain area should be considered because of the fine regulation of this network.

Changes in firing and excitability of the inhibitory system initiate a cascade of inter-cellular events with several electrical but also neurochemical consequences in the perilesional survived tissue. The alteration of GABA release after a stroke can strongly impact the activity and the expression of GABA-A receptors that are the main mediators of the GABAergic signalling. The consequence is an altered modulation of both tonic and phasic inhibition that was shown to have different consequences on the functional outcomes. Phasic inhibition is due to fast activation of postsynaptic GABA-A receptors in response to release of GABA in the synaptic cleft. GABA-A receptors involved in phasic inhibition contain α1, α2 or α3 subunits. On the other hand, GABA molecules which leave the synaptic cleft are able to modulate membrane potential by activating other subtypes or GABA-A receptors containing α4 or α5 subunits. These differences in receptors allow a specific pharmacological modulation of phasic and tonic inhibition. An ischemic event leads to a consistent reorganization of GABA-A receptors exposed by cell membranes [150] creating a highly complex picture that is just starting to be decoded. A study from Hiu et al. reported an increased phasic inhibition in the peri-lesional layer V during the critical window of cortical plasticity in a mouse model of ischemic injury. Increasing this phasic inhibition with Zolpidem improved motor outcomes in adhesive tape removal and rotating beam [151]. A more recent paper found a selective increase in phasic inhibition following Continuous theta burst stimulation (cTBS) in mice with a photothrombotic lesion in posterior parietal cortex (PPC) and daily treatment with cTBS improved cognitive function in Morris water maze. These finding suggest a positive role of an enhanced phasic inhibition in the post-acute phase of ischemic injury; however, considering other clinical evidence where the administration of Midazolam re-induced and worsened clinical deficits [152], this topic is still debated. On the other hand, recent studies were focused on manipulation of tonic inhibition. Clarkson et al. demonstrated that tonic GABAergic inhibition results increased after stroke and that a negative modulation of this tonic inhibition with a benzodiazepine inverse agonist improved functional outcomes in mice after ischemic injury [125,153]. Accordingly, we found that a downregulation of GABA presynaptic signaling in the first week post-stroke in mice, significantly improved general motor function with a long term effect [154]. In the same line, in another study on tMCAo in mice showed that stroke-induced glutamate release activates NMDA receptors, thereby reducing KCC2 transporters and down-regulating extrasynaptic GABA-A receptors. Functionally, this was associated with improved motor performance on the RotaRod motor test. However, as an adverse side effect, decreased tonic inhibition facilitates post-stroke epileptic seizures [155]. Moreover, a recent international, randomized, double-blinded clinical study (phase 2) involving the drug S44819, a selective GABAA α5 receptor antagonist which was demonstrated to have a positive effect on stroke recovery in rodents, found no significant improvement on modified Rankin Scale (mRS) and NIHSS and Montreal Cognitive Assessment (MoCA) scores [156]. Regulation of GABAergic activity to improve functional outcomes after ischemic injury requires highly controlled and specific knowledge and interventions, and well targeted studies are needed to unveil the mechanism underlying its role in post-stroke recovery.

## 7. Conclusions

Ischemic stroke is a highly complex pathology and its consequences reverberate over several cellular mechanisms, both aside and distant to the injury core. In this review, we summarized the main categories of inter-cellular alterations induced by the ischemic event. All this literature demonstrates that the ischemic damage alters cell-to-cell chemical and structural signaling, affecting, in a specific manner, excitatory and inhibitory neurons as well as the different types of glial cells (Figure 1). This knowledge represents the starting point for the validation of highly-targeted therapeutic strategies for a more complete recovery of functional outcome after ischemic damage. In this review we mentioned several possible therapeutic approaches addressing cellular and molecular targets involved in cell-to-cell interactions (NogoA, BM-MSC transplantation, modulation of the GABAergic tone, Non-Invasive Brain Stimulation techniques). But many others are in the research spotlight, based on highly innovative bioengineering and genetic techniques like neural reprogramming of glial cells [157], humanized self-organized models [158], organoids as promising tools for transplantation of stem cells or drugs [159], 3D cultures and human microvessel-on-a-chip platforms [160] which represent a powerful tool for in-vitro studies of the human Blood-Brain Barrier (BBB) and some of which can be subjected to high resolution imaging techniques thus allowing real-time monitoring of BBB penetration [161], endothelial activation and leukocyte adherence even in case of a stroke event. All of these novel technologies will pave the way for novel and highly specific therapies to treat stroke patients.

## Figures and Tables

**Figure 1 cells-10-03050-f001:**
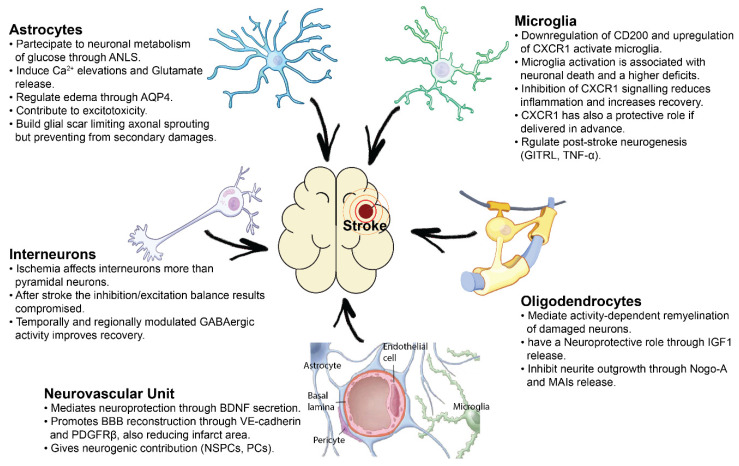
Cell-to-cell interactions after cerebral ischemia. Astrocytes, Micoglia, Oligodendrocytes, Interneurons and Neurovascular Unit are deeply involved in post-stroke plasticity, neuroprotection and brain repair. Abbreviations: blood brain barrier (BBB); brain derivered neurotrophic Factor (BDNF), pericytes (PCs), neural stem progenitor cells (NSPCs), platelet derived growth factor receptor β (PDGFRβ), insulin-like growth factor-1 (IGF-1), myelin associated inhibitors (MAIs), astrocyte-neuron lactate shuttle (ANLS), aquaporin 4 (AQP4), Tumor Necrosis factor α (TNF-α), Glucocorticoid-induced tumor necrosis factor receptor family-related protein ligand (GITRL).

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
