# Peer review of "Cell-to-Cell Interactions Mediating Functional Recovery after Stroke"

_cells, 2021, doi:10.3390/cells10113050_

Round 1
Reviewer 1 Report
This paper meticulously reviewed the pathophysiological role of astrocytes, oligodendrocytes, microglia, and neuron, etc. I read this paper with great interest.
Although the role of individual cell types is important, concepts of the neurovascular unit have been evolved according to the newly disclosed role of the individual cell types. I strongly recommend adding a new section presenting an overview for a holistic interaction among different cell types in the milieu of NVU or oligovascular niche, etc.
Some mistype
L159 oligondendrycytes
Author Response
Although the role of individual cell types is important, concepts of the neurovascular unit have been evolved according to the newly disclosed role of the individual cell types. I strongly recommend adding a new section presenting an overview for a holistic interaction among different cell types in the milieu of NVU or oligovascular niche, etc.
Response: We thank the Reviewer for this precious suggestion; we inserted the new section as recommended as paragraph “Cellular and molecular support from the Neurovascular unit after ischemia” (lines 72-126). We fixed typos.
Reviewer 2 Report
In this review article, authors described the various types of cells, including astrocyte, oligodendrocytes, OPCs, neurons, and microglia mainly by focusing on their cell-cell interaction under ischemic stroke. I think this review article is interesting and well written. However, to improve the quality of this article, I listed the following items.
- It is better to include some schematic illustration on the main points of current paper in order to one can easily understand the content of this article.
- Authors described on roles of neurons, glial cells (astrocytes, oligodendrocytes), and microglia after ischemic stroke. However, besides of these cells, brains consist of endothelial cells and pericytes. Recent studies showed that a part of endothelial cells and pericytes could survive even after ischemic insult (Cells, 9, 1374, 2020; International Journal of Molecular Sciences, 21, 6360, 2020) and that the traits of endothelial cells and pericytes altered following ischemic stroke (Stem Cells, 33, 1962-1974, 2015; Journal of Neuroinflammation, 13, 57, 2016; Histology and Histopathology, 33, 507-521, 2018). In addition, besides OPCs, it is well known that several types of stem/progenitor cells emerge in the brains, especially under ischemic condition (Translational Stroke Research, 8, 515–528, 2017). The fate of stem cells was influenced by surrounding cells, such as endothelial cells (Stem Cells, 27, 2185-2195, 2009; Stem Cells, 28, 1292-1302, 2010) and inflammatory cells (Cell death and Differentiation, 19, 756-767, 2012) through cell-cell contact. The discussion about above mentioned issues by citing some papers (if necessary) would enrich the content of this paper and make wide range field of reader more attractive paper.
Minor:
P2, Line 87
“n this way” should be corrected “In this way”.
Author Response
- It is better to include some schematic illustration on the main points of current paper in order to one can easily understand the content of this article.
Response: We totally agree with the Reviewer and we now included Figure 1 representing a schematic resume of all the main points of the review to facilitate the reader.
- Authors described on roles of neurons, glial cells (astrocytes, oligodendrocytes), and microglia after ischemic stroke. However, besides of these cells, brains consist of endothelial cells and pericytes. Recent studies showed that a part of endothelial cells and pericytes could survive even after ischemic insult (Cells, 9, 1374, 2020; International Journal of Molecular Sciences, 21, 6360, 2020) and that the traits of endothelial cells and pericytes altered following ischemic stroke (Stem Cells, 33, 1962-1974, 2015; Journal of Neuroinflammation, 13, 57, 2016; Histology and Histopathology, 33, 507-521, 2018). In addition, besides OPCs, it is well known that several types of stem/progenitor cells emerge in the brains, especially under ischemic condition (Translational Stroke Research, 8, 515–528, 2017). The fate of stem cells was influenced by surrounding cells, such as endothelial cells (Stem Cells, 27, 2185-2195, 2009; Stem Cells, 28, 1292-1302, 2010) and inflammatory cells (Cell death and Differentiation, 19, 756-767, 2012) through cell-cell contact. The discussion about above mentioned issues by citing some papers (if necessary) would enrich the content of this paper and make wide range field of reader more attractive paper.
Response: We really appreciated this suggestion and, in line with the comment of Reviewer 1, we inserted a novel paragraph “Cellular and molecular support from the Neurovascular unit after ischemia” (lines 72-126), where we addressed all of these points, together with a part in the section “Microglial-mediated inflammation in stroke: a double-edged sword” (lines 346-376). We fixed typos.
Reviewer 3 Report
The authors review the literature on cell‐to‐cell interactions mediating functional recovery after stroke. The review is of potential interest; however, it has significant shortcoming that would need to be addressed.
The sections for each cell should include a figure and table with the corresponding literature. In addition, the overall idea, as to regard to the logistic in understanding the topic, has to be presented in a figure, which should also have the rationale for potential treatments.
The entire text has to be extensively revised for proper usage of English. In the abstract the authors write: “Holy Graal” It is Holy Grail!!
Author Response
The sections for each cell should include a figure and table with the corresponding literature. In addition, the overall idea, as to regard to the logistic in understanding the topic, has to be presented in a figure, which should also have the rationale for potential treatments.
Response: We agree with the necessity of a figure resuming the main points of the article. In line with the suggestions of the other Reviewers, we now include Figure 1.
The entire text has to be extensively revised for proper usage of English. In the abstract the authors write: “Holy Graal” It is Holy Grail!!
Response: We revised the entire manuscript for typos.
Reviewer 4 Report
The manuscript by Alia et al. discusses the cellular interactions after ischemic injury and the potential therapeutic benefits of targeting these interactions and developing new treatments for stroke.
The review is comprehensive, informative, nicely-written, timely and up-to-date (in most parts). Authors were successful in providing some well compiled opinions and summaries which makes this review a good starting point for researchers interested in stroke among Cells readers and beyond.
However, there is a number of major and minor points that would need to be addressed in order to improve the quality of this paper before it can be accepted for publication.
General:
- This review overlooked some essential and up-to-date work regarding the recent advances in target validation and future therapies. I have made some suggestions below but authors are encouraged to consider citing updated references throughout the review, whenever possible.
Major:
-Line 46-48 “Those interventions may involve immunosuppressive agents [4], mediators of inflammatory response [5] and NMDAR‐mediated NO production [6], but also stem cells [7] and hypothermia [8]”. Authors need to refer to the recent advances in regulating brain water homeostasis and glymphatic clearance as a novel approach for targeting one of the underlying causes for poor prognosis in stroke. References to be added:
https://pubmed.ncbi.nlm.nih.gov/34499128/
https://pubmed.ncbi.nlm.nih.gov/34408336/
-Line 67-69: It’s advised to mention that the brain consumes about 20% of total energy. This will help in explaining “An ischemic stroke is a devastating metabolic shock for brain tissue and potently impacts not only on neuronal populations, but also on the organization of their supporting 68 system” through discussing the ANLS theory. The role of brain energetic to be discussed. Authors need to mention the astrocyte‐neuron lactate shuttle (ANLS) hypothesis postulated in 1994 (Pellerin and Magistretti 1994). According to this, astrocytes serve as a ‘lactate source’ whereas neurons serve as a ‘lactate sink’. Moreover, the opposition by Bak and colleagues who argued that oxidative metabolism of lactate within neurons only occurs during repolarization (and in the period between depolarizations) rather than during neurotransmission activity. The emerging role of astrocytes has helped in settling this debate in favour for ANLS hypothesis. References to be included:
https://pubmed.ncbi.nlm.nih.gov/31318452/
https://pubmed.ncbi.nlm.nih.gov/19393013/
https://pubmed.ncbi.nlm.nih.gov/7938003/
-Section 2 “Pathophysiological role of astrocytes and reactive gliosis after brain ischemia”. The authors omitting a key study from 2020, demonstrating that the development of edema following injury-induced hypoxia is AQP4 dependent. That study shows that ischemia and CNS edema are associated with increases both in total aquaporin-4 expression and aquaporin-4 subcellular translocation to the blood-spinal-cord-barrier (BSCB). Pharmacological inhibition of AQP-4 translocation to the BSCB helped in the treatment of ischemia-induced CNS edema and promotes functional recovery in injured rats.
This role has been recently been confirmed by the work of Sylvain et al BBA 2021 which has demonstrated that targeting astrocytes effectively reduces cerebral edema during the early acute phase in in stroke using photothrombotic stroke model. They have also shown a link to brain energy metabolism as indicated by the increase of glycogen levels. Reference to be included:
https://www.cell.com/cell/fulltext/S0092-8674(20)30330-5.
https://pubmed.ncbi.nlm.nih.gov/33561476/
The authors should include these important publications in their discussion.
Minor:
-Authors need to briefly discuss future directions following towards the end of their discussion and conclusion. This could include, but not limit to, the use of humanized self-organized models, organoids, 3D cultures and human microvessel-on-a-chip platforms especially those which are amenable for advanced imaging such as TEM and expansion microscopy since they enable real-time monitoring of brain penetration, endothelial activation and leukocyte adherence during stroke and related CNS disorders. References to be included:
https://pubmed.ncbi.nlm.nih.gov/30165870/
https://pubmed.ncbi.nlm.nih.gov/33117784/
https://pubmed.ncbi.nlm.nih.gov/31889243/
-Line 14: typo- Holy Grail.
-Line 28-29: “Indeed, the year 2017 saw 1.12 million of 28 stroke episodes in the European Union, with 0.46 million deaths”. Use updated statistics from 2021 or 2020 as the earliest.
Best.
Author Response
-Line 46-48 “Those interventions may involve immunosuppressive agents [4], mediators of inflammatory response [5] and NMDAR‐mediated NO production [6], but also stem cells [7] and hypothermia [8]”. Authors need to refer to the recent advances in regulating brain water homeostasis and glymphatic clearance as a novel approach for targeting one of the underlying causes for poor prognosis in stroke. References to be added:
https://pubmed.ncbi.nlm.nih.gov/34499128/
https://pubmed.ncbi.nlm.nih.gov/34408336/
Response: We thank the Reviewer for this interesting hint. We follow the suggestion and we discussed about novel approaches involving water homeostasis and glymphatic clearance in brain tissue in the introduction (lines 52-54).
-Line 67-69: It’s advised to mention that the brain consumes about 20% of total energy. This will help in explaining “An ischemic stroke is a devastating metabolic shock for brain tissue and potently impacts not only on neuronal populations, but also on the organization of their supporting 68 system” through discussing the ANLS theory. The role of brain energetic to be discussed. Authors need to mention the astrocyte‐neuron lactate shuttle (ANLS) hypothesis postulated in 1994 (Pellerin and Magistretti 1994). According to this, astrocytes serve as a ‘lactate source’ whereas neurons serve as a ‘lactate sink’. Moreover, the opposition by Bak and colleagues who argued that oxidative metabolism of lactate within neurons only occurs during repolarization (and in the period between depolarizations) rather than during neurotransmission activity. The emerging role of astrocytes has helped in settling this debate in favour for ANLS hypothesis. References to be included:
https://pubmed.ncbi.nlm.nih.gov/31318452/
https://pubmed.ncbi.nlm.nih.gov/19393013/
https://pubmed.ncbi.nlm.nih.gov/7938003/
-Section 2 “Pathophysiological role of astrocytes and reactive gliosis after brain ischemia”. The authors omitting a key study from 2020, demonstrating that the development of edema following injury-induced hypoxia is AQP4 dependent. That study shows that ischemia and CNS edema are associated with increases both in total aquaporin-4 expression and aquaporin-4 subcellular translocation to the blood-spinal-cord-barrier (BSCB). Pharmacological inhibition of AQP-4 translocation to the BSCB helped in the treatment of ischemia-induced CNS edema and promotes functional recovery in injured rats.
This role has been recently been confirmed by the work of Sylvain et al BBA 2021 which has demonstrated that targeting astrocytes effectively reduces cerebral edema during the early acute phase in in stroke using photothrombotic stroke model. They have also shown a link to brain energy metabolism as indicated by the increase of glycogen levels. Reference to be included:
https://www.cell.com/cell/fulltext/S0092-8674(20)30330-5
https://pubmed.ncbi.nlm.nih.gov/33561476/
The authors should include these important publications in their discussion.
Response: We really appreciated these important suggestions and reflections and we thank the Reviewer for underlying this lack in our dissertation. We addressed all of these points and inserted the suggested references in the paragraph “Pathophysiological role of astrocytes and reactive gliosis after brain ischemia” (lines 130-146) and (lines 175-192).
Minor:
-Authors need to briefly discuss future directions following towards the end of their discussion and conclusion. This could include, but not limit to, the use of humanized self-organized models, organoids, 3D cultures and human microvessel-on-a-chip platforms especially those which are amenable for advanced imaging such as TEM and expansion microscopy since they enable real-time monitoring of brain penetration, endothelial activation and leukocyte adherence during stroke and related CNS disorders. References to be included:
https://pubmed.ncbi.nlm.nih.gov/30165870/
https://pubmed.ncbi.nlm.nih.gov/33117784/
https://pubmed.ncbi.nlm.nih.gov/31889243/
Response: We agree with the Reviewer about the necessity to extend the point regarding future perspectives related to therapeutic interventions and we included the suggested thematic in the paragraph “Conclusions”, (lines 607-619).
-Line 14: typo- Holy Grail.
Response: We fixed the typo.
-Line 28-29: “Indeed, the year 2017 saw 1.12 million of 28 stroke episodes in the European Union, with 0.46 million deaths”. Use updated statistics from 2021 or 2020 as the earliest.
Response: We now included the 2021 Heart Disease & Stroke Statistical Update from the American Health Association regarding the retrospective statistics of year 2019 (lines 32-34).
Round 2
Reviewer 1 Report
I feel my concerns are fully addressed in the revised manuscript.
Reviewer 3 Report
The authors have revised the manuscript as requested.
Reviewer 4 Report
The authors have successfully addressed the majority of my comments and concerns in order to improve the quality of the manuscript.
I believe that the new sections, improved figure, and updated references, have contributed to enhancing the clarity of the manuscript, which I can now endorse for publication following one minor edit.
All the best!
Minor:
-Line 623-625 " human Blood‐Brain Barrier (BBB) and some of which can be subjected to high-resolution imaging techniques thus allowing real‐time monitoring of BBB penetration". The reference for this particular work is missing which should be (PMID: 33117784).